# In Vitro Synergy of Colistin in Combination with Meropenem or Tigecycline against Carbapenem-Resistant *Acinetobacter baumannii*

**DOI:** 10.3390/antibiotics10070880

**Published:** 2021-07-20

**Authors:** Jacinda C. Abdul-Mutakabbir, Juwon Yim, Logan Nguyen, Philip T. Maassen, Kyle Stamper, Zain Shiekh, Razieh Kebriaei, Ryan K. Shields, Mariana Castanheira, Keith S. Kaye, Michael J. Rybak

**Affiliations:** 1Anti-Infective Research Laboratory, Department of Pharmacy Practice, Eugene Applebaum College of Pharmacy and Health Sciences, Wayne State University, Detroit, MI 48202, USA; jabdulmutakabbir@llu.edu (J.C.A.-M.); juwonyim@yahoo.com (J.Y.); logan.nguyen@wayne.edu (L.N.); philip.maassen@wayne.edu (P.T.M.); Kyle.stamper@wayne.edu (K.S.); Zshiekh@wayne.edu (Z.S.); razieh.kebriaei@wayne.edu (R.K.); 2Division of Infectious Diseases, University of Pittsburgh Medical School, Pittsburgh, PA 15213, USA; shieldsrk@upmc.edu; 3JMI Laboratories, North Liberty, IA 52317, USA; mariana-castanheira@jmilabs.com; 4Division of Infectious Diseases, University of Michigan Medical School, Ann Arbor, MI 48109, USA; keithka@med.umich.edu; 5Department of Pharmacy Services, Detroit Receiving Hospital, Detroit Medical Center, Detroit, MI 48201, USA; 6School of Medicine, Wayne State University, Detroit, MI 48201, USA

**Keywords:** *Acinetobacter baumannii*, multidrug-resistant, carbapenem-resistant, colistin-resistant

## Abstract

*Acinetobacter baumannii* is currently classified as one of six pathogens that contribute to increased patient mortality. Thus, exploratory studies navigating alternative treatment strategies are of supreme interest. Herein, we completed minimum inhibitory concentration (MIC) testing, and time-kill analyses (TKA) on 50 carbapenem-resistant *Acinetobacter*
*baumannii* isolates including 28 colistin-resistant isolates. Upon testing of MEM or TGC in the presence of sub-inhibitory COL against the 50 isolates, there was a median 2-fold reduction in MEM and TGC MICs. In the TKAs, the COL+MEM combination was synergistic in 45 (90%) isolates and bactericidal in 43 (86%) isolates at 24 hours, whereas the COL+TGC combination TKAs demonstrated synergy in 32 (64%) isolates and bactericidal activity was shown in 28 (56%) isolates. Additionally, sulbactam (SUL) and TGC were added to the COL+MEM dual therapy regimen to assess the possible utility of a triple therapy regimen against five non-responsive isolates. The COL+MEM+SUL and COL+MEM+TGC regimens effectively restored synergy in (5/5) 100% of the isolates. The results of this study demonstrate the potential utility of COL combinations in the treatment of carbapenem-resistant isolates.

## 1. Introduction

*Acinetobacter baumannii*, a nonfermenting Gram-negative pathogen, is among the most challenging nosocomial pathogens to eradicate [1,2]. Commonly isolated in a spectrum of hospital-associated diseases including ventilator-associated pneumonia, bacteremia, urinary tract infections, and wound infections, the multiple inherent and acquired antibiotic resistance mechanisms possessed by *A. baumannii* increases the likelihood of failure of therapy with antimicrobials active against Gram-negative bacteria [3,4]. Carbapenem-resistant *A. baumannii* isolates have been increasingly reported in the past decade, contributing to a rise in treatment failure [5]. 

In this context, carbapenems have been readily utilized as the most appropriate agents in successfully treating antimicrobial resistant *A. baumannii* infections; however, a worldwide surge in carbapenem resistance has limited their effectiveness [5,6,7]. To combat carbapenem-resistant A. *baumannii*, sulbactam, a beta-lactamase inhibitor typically used in combination with ampicillin, has been employed for its inherent activity against *Acinetobacter* spp. Similar to the carbapenems, resistance rates to sulbactam have risen to greater than 40%, limiting its efficacy in eradicating highly resistant *A. baumannii* infections [8]. 

Amid the propagation of carbapenem-resistant *A. baumannii*, colistin (COL) has been consistently utilized as salvage therapy, as most *A. baumannii* isolates retain susceptibility to this agent [9,10]. Nevertheless, monotherapy with COL often results in sub-optimal clinical outcomes attributed to ill-defined dosing parameters, as well as nephrotoxicity [11,12]. 

Due to the absence of viable alternatives, it has been proposed that COL be used in combination with other antimicrobial agents, although to date, there are very little clinical data supporting this recommendation. The mechanism defining COL’s ability to decrease other antimicrobials MICs against *A. baumannii* when used in combination therapy remains unclear; however, COL has been described to disrupt the integrity of the Gram-negative outer membrane [3,13]. This disruption of the membrane may have a positive impact on a number of antibiotics, increasing their permeability and ultimately allowing them to exhibit improved activity [14]. Various COL combinations have been explored, including those that involve COL in combination with carbapenems, aminoglycosides, and tetracyclines with *A. baumannii* activity [13,15,16,17]. Despite these studies, the most effective combinations are yet to be determined due to the scarcity of bacterial isolates collected, different methods for synergy assessment, and ultimately the inability to associate in vitro study results with clinical outcomes. 

The objective of this study was to investigate combination COL combinations with meropenem (MEM) or tigecycline (TGC) against carbapenem-resistant, including COL-resistant, *A. baumannii* isolates via broth microdilution susceptibility testing and time-kill analysis.

## 2. Results

All *A. baumannii* isolates were resistant to MEM (MICs ranging from ≥8 to ≥64 μg/mL) and 28 isolates were resistant to COL (COL MICs ranging from 4 to 256 μg/mL). Due to the lack of clinical laboratory and standards institute (CLSI) breakpoints for TGC and *Acinetobacter baumannii*, susceptibility rates for TGC could not be inferred; nevertheless, our MIC_90_ was 4 mg/L (TGC values ranged from 0.25 to 8 mg/L), comparable to various studies in which TGC MIC values of 4 or greater were reported [18,19]. Overall MEM or TGC plus sub-inhibitory COL reduced MIC values a median of twofold (range up to 512-fold) from the baseline. We observed the same twofold reduction (range up to 64-fold) for the 28 COL-resistant isolates with these combinations. The MIC_50_ and MIC_90_ values of COL, MEM, and TGC alone, as well as MEM and TGC in the presence of COL, are summarized in Table 1. 

In the TKA, the combination of COL+MEM demonstrated synergy in 45 of 50 isolates (90%) and bactericidal activity at 24 hours in 43 of 50 isolates (86%) (Appendix A); while the COL+TGC combination demonstrated synergy in 32 of 50 isolates (64%) and bactericidal activity in 28 of 50 (56%) isolates. Of interest, 27/28 (96%) COL-resistant isolates tested showed synergy with the COL+MEM regimen, while 23/28 (79%) of the isolates showed synergy with the COL+TGC combination. The presence of synergy was not influenced by the elevated single MICs of COL, MEM, or TGC. Figure 1 shows the TKAs of three included isolates, with varying COL susceptibility. The majority of the TKAs demonstrated synergy with the COL+MEM combination. 

Noting the high degree of synergy shown with the COL+MEM combination, with only five isolates not responsive to the regimen, additional TKAs were completed against these five isolates with the inclusion of SUL or TGC into a three-drug regimen (COL+MEM+SUL, COL+MEM+TGC). The COL+MEM+SUL and COL+MEM+TGC triple combinations were both synergistic and bactericidal against each of the five isolates. The COL+MEM+SUL therapy produced an average 5.15 reduction in CFU/ml from the most active single agent compared to the average 5.29 reduction in CFU/ml from the most active single agent observed with the COL+MEM+TGC therapy. A graphical representation of the decline in CFU/ml with the use of the triple combinations (COL+MEM+SUL, COL+MEM+TGC) is depicted in Figure 2.

Further, we completed whole genome sequencing and transcriptome analysis in four of the five of these isolates that did not demonstrate synergy with the COL+MEM combination (the fifth strain was not available for molecular analysis). The isolates belonged to the international clones ST2 and ST3. All isolates harbored the acquired OXA-23 Class D beta-lactamase, and this gene was overexpressed (>50× compared to the control isolates) in one isolate. Different variants of the intrinsic OXA-51 Class D beta-lactamase (shown as OXA-71 and OXA-66) and the AmpC beta-lactamases, variants of ADC (*A. baumannii* specific chromosomal cephalosporinases), were also detected. ADC was overexpressed in two isolates. The gene encoding TEM-1 was observed in one isolate.

The analysis of the differential expression of efflux pump components and its regulators and outer membrane proteins revealed decreased expression of OmpA (<10× in all four isolates; two also had decreased expression of CarO and one had decreased expression of OprD. The impaired expression of OmpA [20] and CarO has been correlated with increases in carbapenem MICs [21]. The two isolates with decreased CarO expression had a S119T substitution at the polymorphic site in PmrA which has been linked to COL resistance [22]. The results of the whole genome sequencing and transcriptome (RNA-seq) analysis are shown in Table 2.

## 3. Discussion

Our data demonstrate a number of key findings. First and foremost, we showed that irrespective of elevated MICs to COL, MEM, or TGC, COL-based combination regimens are synergistic against carbapenem-resistant A. baumannii isolates. Second, we showed that sub-inhibitory amounts of COL were efficacious in decreasing TGC and MEM MICs. As all of the tested isolates were resistant to MEM, MICs ≥ 8 mg/L and 90% of the isolates had TGC MICs > 4 mg/L; current literature suggests that these elevated MICs to either MEM or TGC would dampen their clinical efficacy when utilized as a monotherapy option [23,24,25]. With that, it is important to note the potential impact that COL could have on lowering the MIC of either MEM or TGC. Similarly, in isolates that were both COL and carbapenem-resistant isolates, we demonstrated that sub-inhibitory amounts of MEM and TGC were able to decrease COL MICs. 

Despite the positive results observed, it was important to explore the isolates with a lack in response to the COL+MEM dual therapy. Notably, the isolates had a multitude of resistance mechanisms, all likely contributors to the decreased response observed with the combination. The OXA-23 carbapenamase, specifically, has been commonly identified as a major mechanism in carbapenem resistance observed in *A. baumannii* isolates [26,27]. It has been suggested that under selective pressure with a carbapenem agent, organisms are able to not only overexpress OXA-23 but also acquire additional mechanisms of resistance, such as modifications to porin channels and the overexpression of extended spectrum beta-lactamases, as seen in our study [28]. 

Furthermore, in *A. baumannii*, COL resistance occurs through two specific mechanisms: the enzymatic modification of the lipid A moieties of the bacterial cell wall, or by the complete loss of cell wall lipopolysaccharide (LPS) [29]. COL resistance caused by either mechanism is presumed to be mediated by deletions or substitutions in the PmrAB system [29]. Although we did not observe substitutions in PmrB, we did note that two of the isolates that underwent molecular analysis due to lack in synergy to the COL+MEM combination did have the S119T substitution in PmrA at the polymorphic site. 

Previous investigators observed synergistic activity with the COL+carbapenem+SUL triple therapy regimen and hypothesized that the improved killing effects may have been attributed to the increased affinity that both carbapenems and sulbactam have to penicillin-binding protein (PBP2) and PBP3 [17]. Similar to the success shown in that study, we observed increased activity with the addition of SUL to the COL+MEM regimen when tested in the five isolates that were nonresponsive to the COL+MEM combination therapy, thus building upon the initial hypothesis that the increased occupancy of the PBPs present within the *A. baumannii* isolates equates to increased killing activity. Further, it is possible that the enhanced membrane permeabilization exhibited by COL plays a role in facilitating the binding of SUL and MEM to their binding sites, despite the resistance mechanisms noted in the organisms [30,31]. 

Additionally, both MEM and COL have been shown to be synergistic when used in combination with TGC (MEM+TGC, and TGC+COL) [32,33]. As it has been identified throughout the literature that the use of multiple agents with differing mechanisms of action is efficacious in overcoming increased resistance, we also tested TGC in combination with COL+MEM against the five isolates that were not responsive to the COL+MEM combination regimen. Once again, we observed increased activity with the COL+MEM+TGC triple therapy regimen when tested against the five nonresponsive isolates. 

Of interest, the first randomized trial comparing COL monotherapy and COL in combination with MEM against carbapenem-resistant Gram-negative bacteria, including *A. baumannii*; did not demonstrate a significant reduction in patient failures when compared to COL alone (73%, COL+MEM, vs 79% COL, *p*-value = 0.172). Of note, all patients treated in this study had COL susceptible isolates at baseline and therefore, the impact of this combination on COL and carbapenem-resistant *A. baumannii* was not evaluated [34]. Further, a similarly designed multi-center, multi-country randomized trial in progress is also evaluating the combination of MEM and COL for carbapenem-resistant Gram-negative infections. This study may supply additional information on the usefulness of this combination, clinically [35].

While both in vitro and in vivo research of the COL+TGC combination is limited, similar studies have shown comparable results to our study with the COL+TGC combination [36,37]. Referring back to the five isolates that were not responsive to the COL+MEM combination, four of those isolates did present with synergy when tested against the COL+TGC combination regimen, thus showing that the COL+TGC combination is capable of producing synergy against isolates that are not responsive to the COL+MEM combination.

Our study did encounter similar limitations described by the aforementioned studies regarding the evaluation of COL combination therapy. A small number of included isolates were investigated for detailed phenotypes and genotypes regarding specific mechanisms for COL, MEM, TGC resistance or synergy which may limit the application of these findings to clinical settings. Additionally, we were unable to complete core-genome MLST which may have served as a method to differentiate the *A. baumannii* clones present within the isolates utilized in the study. In addition, the TKA, and combination MIC testing are short-duration experiments using static concentrations, which differs from humanized pharmacokinetic concentration exposure conditions that can be mimicked in animal and in vitro pharmacokinetic and pharmacodynamic models. 

## 4. Materials and Methods

### 4.1. Bacterial Isolates

A total of 50 carbapenem-resistant *A. baumannii* unique clinical isolates were evaluated in this study. A total of 18 isolates were collected from institutions in the United States, 14 were collected from Thailand institutions, and the remaining 18 isolates were collected from institutions located in Israel. Carbapenem resistance was defined as isolates that had a MEM MIC ≥ 8 mg/L. Of the 50 evaluated isolates, 28 of the isolates were COL-resistant. As there is a gap in the available literature discussing the treatment of COL-resistant infections and the use of combination antimicrobial regimens, the 28 COL-resistant isolates were intentionally included in the collection of tested isolates. The isolates were collected from patients that were part of an NIH-funded clinical trial studying pneumonia and/or bloodstream infection due to extremely drug-resistant Gram-negative pathogens (NCT01597973) [35]. In an effort to further present resistance mechanisms, whole genome sequencing and transcriptome analysis was completed on four of the five isolates (R9314, R9751, R9761, R11542) shown to be non-responsive to the COL+MEM combination.

#### Antimicrobials

COL, MEM, and SUL were purchased commercially from Sigma Chemical Co. (St. Louis, MO, USA), and TGC was obtained from its manufacturer (Pfizer^®^ New York, NY, USA).

### 4.2. Susceptibility Testing

Susceptibility testing for COL, MEM, and TGC was performed for each strain in 96-well microtiter plates (Corning^®^ Costar^®^, obtained though Sigma-Aldrich^®^, Warren, MI, USA) by broth micro-dilution using cation-adjusted Mueller–Hinton broth (CAMHB; Difco, Detroit, MI, USA) supplemented with 25 mg/L Ca^2+^ and 12.5 µg/ml Mg^2+^ according to the Clinical and Laboratory Standards Institute (CLSI) guidelines [23]. Freshly prepared Mueller–Hinton broth was used to prevent the oxidative degradation of TGC in aqueous solution [38]. MIC plates were incubated at 37 °C for 18–24 h prior to recording the results. MIC reductions were measured by serial 2-fold dilutions. *Escherichia coli* ATCC 25922 was used as the internal quality control strain. The following was determined for each tested antimicrobial: MIC, MIC_50_, and MIC_90_. In addition, MIC testing with COL in the presence of sub-inhibitory amounts of MEM or TGC (using CAMHB containing 0.5× MIC or the biological free peak concentration of MEM or TGC, whichever was lesser) was performed for each strain. Similarly, MIC testing with MEM or TGC in the presence of sub-inhibitory amounts of COL (using CAMHB containing 0.5× MIC or the biological free peak concentration of COL, 2 mg/L whichever was lesser) was conducted on each strain [39,40]. For reference, the biological free peak concentration is the peak serum concentration, unbound to protein. This concentration is utilized as a surrogate marker for an achievable concentration based upon an elevated minimum inhibitory concentration. 

### 4.3. In Vitro Time-Kill Analysis

Time-kill analyses (TKA) were performed for all 50 isolates using 24-well tissue culture plates. Briefly, macro-dilution TKA experiments were performed in duplicate to target an initial inoculum of ~10^6^ CFU/mL. Each well was treated with no drug, COL, MEM, TGC, COL+MEM, or COL+TGC, at a concentration of 0.5× the MIC or the biological free peak concentration (MEM *f*Cmax 30 mg/L per 1 g q 8 h dosing, COL *f*Cmax 2 mg/L per 3 million IU q 8 h dosing, TGC *f*Cmax 1.5 mg/L per 100 mg loading dose followed by 50 mg q 12 h dosing) [39,40,41,42], whichever was lower. Additionally, the five isolates that did not respond to the COL+MEM combination were tested in triple drug combinations which included wells treated with COL+MEM+SUL or COL+MEM+TGC, with each drug given at 0.5× the MIC or the biological free peak concentration (whichever was lesser). Experiments were conducted in a shaker incubator at 37 °C for 24 h, and aliquots of 0.1 mL were obtained from each well at 0, 4, 8, and 24 h. Each sample was then diluted in 0.9% normal saline to the appropriate bacterial concentrations, plated on TSA plates using an automated spiral platter (EasySpiral Pro^®^ Intersciences, Woburn, MA, USA), and incubated at 37 °C for 24 h before colony enumeration using an automated colony counter (Scan 1200, Interscience Laboratories Inc., Woburn, MA, USA). Time-kill curves were generated by plotting mean colony counts remaining from duplicate experiments against each time point using Prism^®^ (Graphpad Software, San Diego, CA, USA). The synergistic effect of a combination was defined as both a ≥2-log_10_ reduction in colony forming units (CFU)/mL from the most active single agent at 24 h and a ≥2-log_10_ CFU/mL reduction from the initial inoculum when using the combination. Bactericidal activity was defined as ≥3-log_10_ CFU/mL reduction in the bacterial count at 24 h from the starting inoculum. Antagonism was defined as >2-log_10_ CFU/mL increase in bacterial growth in 24 h from the starting bacterial inoculum. 

### 4.4. Whole Genome Sequencing Analysis of Antimicrobial Resistance Genes

Total genomic DNA was extracted and used as input material for the library construction. DNA libraries were prepared using the Nextera XT™ library construction protocol and index kit (Illumina, San Diego, CA, USA) and sequenced on a MiSeq Sequencer (Illumina). Sequencing analysis was performed after de novo assembly and specific matches were generated for each sample with criteria of >94% identity and 40% minimum coverage length. Mutations were considered present when >50% of the sequence reads allowed for base calling. Sequences displaying 100.0% homology with the reference sequences were named according to the reference. Genes with homology <100.0% were named with suffix “-like” after the gene showing the closest homology. Intrinsic genes were annotated relative to a susceptible reference. Annotations were interpreted by one of the following: wildtype (sequence identical to *A. baumannii* ATCC 17978), alterations (single amino acids substitutions relative to ATCC 17978), or disruptions (alterations resulting in the early termination of a gene or an insertion/deletion of ≥3 continuous amino acids).

### 4.5. Whole Transcriptome Analysis

Total RNA was extracted and purified from log phase bacterial cultures displaying cell density at OD600 of 0.3 to 0.5 using the RNeasy Mini Kit in the Qiacube workstation (Qiagen, Hilden, Germany) according to manufacturer instructions. Residual DNA was eliminated by treatment with RNAse-free DNase (Promega, Madison, WI, USA). Quantification of RNA and sample quality was assessed using the RNA 6000 Pico kit on the Agilent 2100 Bioanalyzer (Agilent Technologies, Santa Clara, CA, USA) according to manufacturer instructions. Only preparations with acceptable RNA integrity numbers (RIN) ≥7 and/or that showed no visual degradation were used for experiments.

A total of up to 2 μg of RNA was subjected to rRNA depletion using Ribo-Zero^®^ (Gram-Negative) rRNA Removal Kit (Illumina) according to the manufacturer’s instructions. Ribo-Zero-treated RNA was purified using the modified RNeasy MinElute option described within the Ribo-Zero protocol and eluted in 14 mL RNase-free water. Eluted samples were again evaluated for quality and quantity using RNA 6000 Pico kit on the Agilent 2100 Bioanalyzer and used same day for library preparation, stored overnight at −20 °C or −80 °C for up to 30 days. Whole transcriptome RNA-Seq cDNA library preparation was performed using the TruSeqTM Stranded mRNA Library Prep (Illumina) with eluted Ribo-Zero-treated RNA samples described above as input material. Library preparation was performed according to the manufacturer’s instructions beginning with fragmentation of mRNA since the depletion of rRNA is done in lieu of the purification step. Fragmentation of mRNA was accomplished by using the entire eluted Ribo-Zero-treated RNA sample (~13 mL) combined with 13 mL of FPF (Fragmentation, Prime, Finish Mix). Sequencing was carried out on MiSeq sequencers using MiSeq Reagent Kit V3 (150-cycle). An independently prepared replicate of the control reference isolate (*A. baumannii* ATCC 17978) was included with each sequencing run to serve as an internal control for all phases of this experiment.

Differential gene expression was evaluated by first pre-processing paired-end FASTQ files using fastp with default parameters to perform quality control, adapter trimming, quality filtering, and per-read quality pruning. The resulting reads were passed to EDGE-pro, which uses Bowtie2 to align reads against the *A. baumannii* ATCC 17978 using default parameters. Mapped reads were filtered based on alignment scores and passed to differential gene expression analysis using the edgeR package. Reads were normalized across samples using trimmed mean of M-values (TMM) normalization and fold change of expression was calculated to an exact test based on a quantile-adjusted conditional maximum likelihood (qCML) method. Gene synonyms and gene ontology (GO) terms were collected from UniProt to aid in interpretation of generated heatmaps, PCA plots, and raw data.

## 5. Conclusions

Overall, we observed synergistic activity with the combination of COL with MEM or TGC in carbapenem-resistant *A. baumannii* including COL-resistant isolates. This information provides support for these combinations as alternative treatment strategies for patients with difficult to treat carbapenem-resistant *A. baumannii* infections. Further research is warranted to determine if these combinations can be used clinically as potential treatment options. 

## Figures and Tables

**Figure 1 antibiotics-10-00880-f001:**
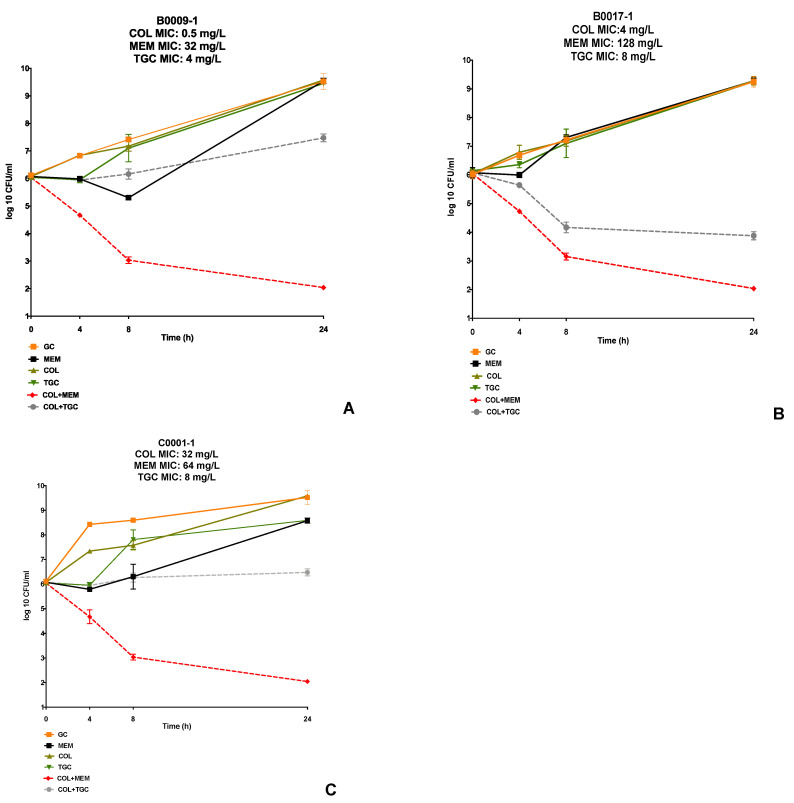
Three representative TKAs, with varying COL-susceptibility (**A**–**C**). GC represents the growth control of each organism tested in each respective TKA. The following concentrations of the antibiotic were utilized in the time-kill experiments: COL was tested at 0.5× the MIC (0.25 mg/L) in B0009-1, and at the biological free peak concertation (2 mg/L) in B0017-1, and C001-1. MEM and TGC were tested at their biological free peak concentrations (30 mg/L and 1.5 mg/L, respectively) against all isolates. The error bars are representative of the standard deviations of the mean, as each TKA was completed in duplicate.

**Figure 2 antibiotics-10-00880-f002:**
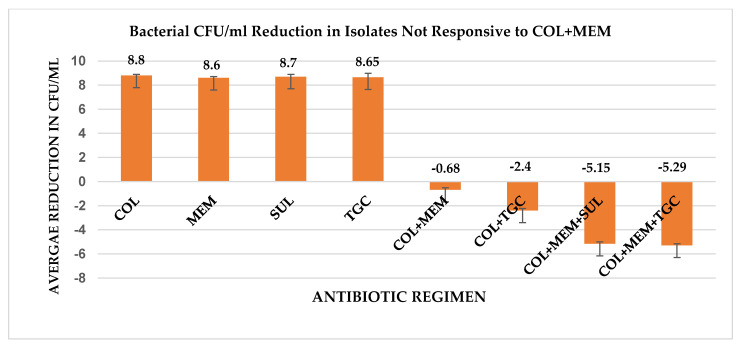
Graphical representation of the decline in bacterial CFU/ml with the use of the triple combinations in the 5 isolates that did not respond to COL+MEM. All TKA experiments, for each respective isolates, were completed in duplicate, and the error bars represent the standard deviation of the mean.

**Table 1 antibiotics-10-00880-t001:** MIC, MIC_50_, and MIC_90_ for COL, MEM, and TGC as collected from single MIC testing by broth microdilution for 50 isolates.

Antimicrobial	MIC Range (mg/L)	MIC_50_ (mg/L)	MIC_90_ (mg/L)	Median Fold-Reduction
COL	0.5–256	4	8	
MEM	8–128	32	128	
TGC	0.25–8	4	4	
MEM+COL	1–64	16	32	2
TGC+COL	0.25–8	2	4	2

MIC, minimum inhibitory concentration; COL, colistin; MEM, meropenem; TGC, tigecycline; MEM+COL, MEM in the presence of COL at 0.5 × MIC or biological free peak concentration; TGC+COL, TGC in the presence of COL at 0.5 × MIC or biological free peak concentration.

**Table 2 antibiotics-10-00880-t002:** Whole genome sequence results for 4 of the 5 isolates that were not responsive to the COL+MEM combination therapy.

Isolate Number	MLST	COL MIC (mg/L)	MEM MIC (mg/L)	Beta-Lactamase Genes	Sequencing Annotation		Differential Expression Analysis (Fold Increase/Decrease in Expression Compared to *A. baumannii* ATCC 17978) ^a^				
					PmrA	PmrB	ADC	OXA (Intrinsic)	OprD	CarO (porin)	OmpA Family
R9314	3	1	16	ADC-79, OXA-23, OXA-71	S119T	P360Q, N440H	12.13	3.65	0.459	0.025	0.154
R9751	3	2	16	ADC-79, OXA-23, OXA-71	A14T, S119T	P360Q, N440H	12.46	5.20	0.201	0.020	0.188
R9761	2	0.5	16	ADC-73, OXA-23, OXA-66, TEM-1	WT	A138T, N440H, A444V	4.35	2.35	0.283	0.594	0.090
R11542	2	4	64	ADC-33, OXA-23, OXA-82	WT	N440H, A444V	3.29	54.08	0.045	0.731	0.085

^a^ Results deemed significant (10× decrease/increase compared to the control isolate) are bolded. Differences were not detected in the expression of the genes encoding the following: AdeS, AdeR, AdeA, AdeB, AdeC, AdeI, AdeJ, AdeK, AdeL, AdeF, AdeG, AdeH, Omp38, and OmpA/MotB. The fifth strain was not available for molecular analysis.

## Data Availability

The data presented in this study are available from authors.

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
