# Peer review of "In Vitro Synergy of Colistin in Combination with Meropenem or Tigecycline against Carbapenem-Resistant *Acinetobacter baumannii"

_antibiotics, 2021, doi:10.3390/antibiotics10070880_

Round 1

Reviewer 1 Report

The article assess the activity of colistin (COL) in combination with meropenem (MEM) or tigecycline (TGC) against carbapenem-resistant Acinetobacter baumannii isolates.  The study demonstrates potential utility of COL combinations in the treatment of carbapenem-resistant isolates. 

Results section.

Some abbreviations should be explained when fist use (e.g., CLSI, ).

References

Maybe it will be useful to include as reference: 

"1542. The Evaluation of the In Vitro Synergy of Colistin in Combination with Meropenem and Tigecycline against 50 Multi-Drug-resistant Acinetobacter baumannii strains"

Author Response

Results section.

Some abbreviations should be explained when fist use (e.g., CLSI, ).

Thank you to the reviewer for this comment, we have defined CLSI in the manuscript.

References

Maybe it will be useful to include as reference: 

 "1542. The Evaluation of the In Vitro Synergy of Colistin in Combination with Meropenem and Tigecycline against 50 Multi-Drug-resistant Acinetobacter baumannii strains"

Thank you to the reviewer for this comment. This reference was not included as the present manuscript is the complete presentation of the results mentioned in this abstract

Reviewer 2 Report

The subject is according to the scope of the Journal. The chosen topic is of scientific interest and the use of English style and grammar is appropriate.

Major point:

This type of laboratory work to obtain the MIC in susceptibility testing and the TKA requires 3 independent experiments. Then, I encourage the Authors to add more experimental work.

Other points:

The abstract should be less descriptive. I advise the Authors to write a brief overview of their research.

To emphasize the importance of Acinetobacter baumannii, I suggest the Authors to add the ESKAPE notion/information to the introduction.

M&M section: It is not stated for which isolates and the reason, WGS, and WGA were performed. Did the Authors utilize one of the isolates responsive to the COL-MEM combination to compare with those nonresponsive, in addition to the reference ATCC17978? This should be useful to better understand the potential resistance mechanisms.

I did not catch, after WGS and WGA studies, the hypothesized potential resistance mechanisms of the isolates not responsive to the COL-MEM.

The conclusion is required.

Minor points:

Line 40: That phrase is not related to the year 2021, then, it should be corrected and clarified. The Authors wrote the manuscript in the year 2013.

Line 79: Please, change the word organism to microorganism, bacteria, or isolates.

4.3: The word strain is not correct in these circumstances. Please, change to “isolate”.

Line 177: Please, describe “PBP2 and PBP3”.  Line 174: Please, describe “S119T” and “PmrB”. Not all the readers know about PBP, S119T and PmrB.

Line 207: “Thus, showing that despite the increase in synergy shown with COL+MEM versus COL+TGC combination regimens against the total number of tested isolates;” Please, rephrase it.

Line 240: Could the Authors clarify what is NCT01597973? Does the manuscript number 40 regard the cited NIH-founded clinical study?

Author Response

This type of laboratory work to obtain the MIC in susceptibility testing and the TKA requires 3 independent experiments. Then, I encourage the Authors to add more experimental work.

We thank the reviewer for this comment. However, the performance of killing curves run in duplicate is fairly standardized for assessing antibiotic activity either alone or in combination against bacterial pathogens.. Killing curves completed in duplicate have been shown to be both accurate and  reproducible. In addition, the performance of MIC testing in duplicate is also standardized and common in the literature with regards to susceptibility testing using microbroth dilution techniques. In this study we have completed a total of 500 MICs and a total of 100 Time-Kill Experiments (done in duplicate w/ growth control). We have included a number of published papers below that have been published using the same exact methods for both MIC and killing curve evaluations. 

Please see the following references:

Mikhail S, Singh NB, Kebriaei R, Rice SA, Stamper KC, Castanheira M, Rybak MJ. Evaluation of the Synergy of Ceftazidime-Avibactam in Combination with Meropenem, Amikacin, Aztreonam, Colistin, or Fosfomycin against Well-Characterized Multidrug-Resistant Klebsiella pneumoniae and Pseudomonas aeruginosa. Antimicrob Agents Chemother. 2019 Jul 25;63(8):e00779-19. doi: 10.1128/AAC.00779-19. PMID: 31182535; PMCID: PMC6658738.

Rose WE, Bienvenida AM, Xiong YQ, Chambers HF, Bayer AS, Ersoy SC. Ability of Bicarbonate Supplementation To Sensitize Selected Methicillin-Resistant Staphylococcus aureus Strains to β-Lactam Antibiotics in an Ex Vivo Simulated Endocardial Vegetation Model. Antimicrob Agents Chemother. 2020;64(3):e02072-19. Published 2020 Feb 21. doi:10.1128/AAC.02072-19

Berti AD, Sakoulas G, Nizet V, Tewhey R, Rose WE. β-Lactam antibiotics targeting PBP1 selectively enhance daptomycin activity against methicillin-resistant Staphylococcus aureus. Antimicrob Agents Chemother. 2013;57(10):5005-5012. doi:10.1128/AAC.00594-13

Shields RK, Nguyen MH, Press EG, Chen L, Kreiswirth BN, Clancy CJ. In Vitro Selection of Meropenem Resistance among Ceftazidime-Avibactam-Resistant, Meropenem-Susceptible Klebsiella pneumoniae Isolates with Variant KPC-3 Carbapenemases. Antimicrob Agents Chemother. 2017 Apr 24;61(5):e00079-17. doi: 10.1128/AAC.00079-17. PMID: 28242667; PMCID: PMC5404588.Perdigão-Neto LV, Oliveira MS, Rizek CF, Carrilho CM, Costa SF, Levin AS. Susceptibility of multiresistant gram-negative bacteria to fosfomycin and performance of different susceptibility testing methods. Antimicrob Agents Chemother. 2014;58(3):1763-7. doi: 10.1128/AAC.02048-13. Epub 2013 Dec 9. PMID: 24323469; PMCID: PMC3957874.

Other points:

The abstract should be less descriptive. I advise the Authors to write a brief overview of their research.

We thank the reviewer for this comment, we have reduced the descriptions presented in the abstract section.

To emphasize the importance of Acinetobacter baumannii, I suggest the Authors to add the ESKAPE notion/information to the introduction.

Thank you to the reviewer for this comment, we have emphasized importance of A. baumannii research.

M&M section: It is not stated for which isolates and the reason, WGS, and WGA were performed. Did the Authors utilize one of the isolates responsive to the COL-MEM combination to compare with those nonresponsive, in addition to the reference ATCC17978? This should be useful to better understand the potential resistance mechanisms.

We thank the reviewer for this comment and have revised the Materials and Methods section to include the following statement, “In an effort to further present resistance mechanisms, whole genome sequencing and transcriptome analysis was completed on four of the five isolates (R9314, R9751, R9761, R11542), shown to be non-responsive to the COL+MEM combination.”

Further, all COL+MEM non-responsive strains were compared to only reference strain to determine potential mechanisms of resistance. This methodology has been utilized throughout literature to determine whether an amplification of resistance genes is present within less susceptible isolates when compared to the reference strains. Additionally, while we could have compared one of the responsive isolates, due to the clones represented amongst A. baumannii and the unknown differences per each isolate, it would be presumptuous to rely on the potential results derived as drivers of resistance

Please see the following references:

Mikhail S, Singh NB, Kebriaei R, Rice SA, Stamper KC, Castanheira M, Rybak MJ. Evaluation of the Synergy of Ceftazidime-Avibactam in Combination with Meropenem, Amikacin, Aztreonam, Colistin, or Fosfomycin against Well-Characterized Multidrug-Resistant Klebsiella pneumoniae and Pseudomonas aeruginosa. Antimicrob Agents Chemother. 2019 Jul 25;63(8):e00779-19. doi: 10.1128/AAC.00779-19. PMID: 31182535; PMCID: PMC6658738.

Abdul-Mutakabbir JC, Nguyen L, Maassen PT, Stamper KC, Kebriaei R, Kaye KS, Castanheira M, Rybak MJ. In Vitro Antibacterial Activity of Cefiderocol against Multidrug-Resistant Acinetobacter baumannii. Antimicrobial Agents and Chemotherapy. 2021 Jun 14:AAC-02646.

Singh, T., Singh, P.K., Das, S. et al. Transcriptome analysis of beta-lactamase genes in diarrheagenic Escherichia coli. Sci Rep 9, 3626 (2019). https://doi.org/10.1038/s41598-019-40279-1

I did not catch, after WGS and WGA studies, the hypothesized potential resistance mechanisms of the isolates not responsive to the COL-MEM.

Thank you to the reviewer for this comment, in the results section, we thoroughly detail the resistance mechanisms represented in the non-responsive strains: “Further, we completed whole genome sequencing and transcriptome analysis in 4 of the 5 of these isolates that did not demonstrate synergy with the COL+MEM combination (the fifth strain was not available for molecular analysis). The isolates belonged to the in-ternational clones ST2 and ST3. All isolates harbored the acquired OXA-23 Class D be-ta-lactamase, and this gene was overexpressed (>50X compared to the control isolates) in 1 isolate. Different variants of the intrinsic OXA-51 Class D beta-lactamase (shown as OXA-71 and OXA-66) and the AmpC beta-lactamases, variants of ADC (A. baumannii specific chromosomal cephalosporinases) were also detected. ADC was overexpressed in 2 isolates. The gene encoding TEM-1 was observed in 1 isolate.The analysis of the differential expression of efflux pump components and its reg-ulators and outer membrane proteins revealed decreased expression of OmpA (<10X) in all 4 isolates; 2 also had decreased expression of CarO and 1 had decreased expression of OprD. The impaired expression of OmpA [22] and CarO has been correlated to increases in carbapenem MICs [23]. The 2 isolates with decreased CarO expression had a S119T substitution at the polymorphic site in PmrA which has been linked to COL-resistance [24]. The results of the whole-genome sequencing and transcriptome (RNA-seq) analysis are shown in Table 2.”

 We then derive hypotheses regarding the potential mechanism of resistance in the discussion section: Notably, the isolates had a multitude of resistance mechanisms, all likely contributors to the decreased response observed with the combination. The OXA-23 carbapenamase, specifically, has been commonly identified as a major mechanism in carbapenem resistance observed in A. baumannii isolates [28,29]. It has been suggested that under selective pressure with a carbapenem agent, organisms are able to not only overexpress OXA-23 but also acquire additional mechanisms of resistance such as: modifications to porin channels, and the overexpression of extended spectrum beta-lactamases, as seen in our study [30]. Furthermore, in A. baumannii, COL-resistance occurs through two specific mechanisms, the enzymatic modification of the lipid A moieties of the bacterial cell wall, or by the complete loss of cell wall lipopolysaccharide (LPS)[31]. COL-resistance caused by either mechanism is presumed to be mediated by deletions or substitutions in the PmrAB system (33). Although we did not observe substitutions in PmrB, we did note that two of the isolates that underwent molecular analysis due to lack in synergy to the COL+MEM combination, did have the S119T substitution in PmrA at the polymorphic site.

The conclusion is required.

Thank you to the reviewer for this comment, the conclusion has been added, as “section 5”.

Minor points:

Line 40: That phrase is not related to the year 2021, then, it should be corrected and clarified. The Authors wrote the manuscript in the year 2013.

We thank the reviewer for this comment, we have updated the section to include a more relevant reference. Please see reference 7.

Line 79: Please, change the word organism to microorganism, bacteria, or isolates.

Thank you to the reviewer for this comment, organism has been changed to isolate.

4.3: The word strain is not correct in these circumstances. Please, change to “isolate”.

We thank the reviewer for this comment, the word “strain” has been changes to isolates throughout.

Line 177: Please, describe “PBP2 and PBP3”.  

Thank you to the reviewer for this comment, penicillin binding protein has been defined.

Line 174: Please, describe “S119T” and “PmrB”. Not all the readers know about PBP, S119T and PmrB. Thank you to the reviewer for this comment, consequently PmrB and S119T are commonly referred to as written, throughout literature. Nevertheless, we did include a line that describes their contribution to A. baumannii resistance. “Furthermore, in A. baumannii, COL-resistance occurs through two specific mechanisms, the enzymatic modification of the lipid A moieties of the bacterial cell wall, or by the complete loss of cell wall lipopolysaccharide (LPS)[31]. COL-resistance caused by either mechanism is presumed to be mediated by deletions or substitutions in the PmrAB system (33). Although we did not observe substitutions in PmrB, we did note that two of the isolates that underwent molecular analysis due to lack in synergy to the COL+MEM combination, did have the S119T substitution in PmrA at the polymorphic site”

Line 207: “Thus, showing that despite the increase in synergy shown with COL+MEM versus COL+TGC combination regimens against the total number of tested isolates;” Please, rephrase it.

We thank the reviewer for this comment, the sentenced has been rephrased, now line: 244-246

Line 240: Could the Authors clarify what is NCT01597973? Does the manuscript number 40 regard the cited NIH-founded clinical study?

Thank you to the reviewer for this comment, “NCT01597973” is the clinical trials number, the reference has been updated to reflect that. Now reference 34.

Reviewer 3 Report

Abdul-Mutakabbir and colleagues have performed an interesting work evaluating the in vitro synergy of colistin in combination with meropenem or tigecycline agains carbapenem-resistant Acinetobacter baumanii.

The paper is well-written and presents important data that should encourage the performance of in vivo studies evaluating these combinations.

Title: Please consider to write Acinetobacter baumanii in italics.

Introduction.

This part is interesting although I found it quite long. Consider if possible to reduce it. 

Line 59 the sentence should be shortened to facilitate its understanding.

Line 77 consider to delete the first term combination. 

Results. 

Figure 1. The figure B is missing. 

Discussion and Methods. OK.

Author Response

Reviewer 3

Abdul-Mutakabbir and colleagues have performed an interesting work evaluating the in vitro synergy of colistin in combination with meropenem or tigecycline against carbapenem-resistant Acinetobacter baumanii.

The paper is well-written and presents important data that should encourage the performance of in vivo studies evaluating these combinations.

Title: Please consider to write Acinetobacter baumannii in italics.

Thank you to the reviewer for this comment, Acinetobacter baumannii is not italicized.

Introduction.

This part is interesting although I found it quite long. Consider if possible to reduce it. 

Thank you to the reviewer for this comment, we have reduced the introductory section.

Line 59 the sentence should be shortened to facilitate its understanding.

Thank you to the reviewer for this comment, the sentence has been reduced. Now lines: 63-64

Line 77 consider to delete the first term combination. 

Thank you to the reviewer for this comment, the word, “combination” has been deleted. Now lines: 119-120

Results.

Figure 1. The figure B is missing. 

Round 2

Reviewer 2 Report

The Authors answered to my questions correctly and amended the manuscript in a proper version. Consequently, I have accepted the manuscript in its updated form.

The use of at least 3 technical and/or 3 biological replicates is highly recommended to improve the quality of the results (minimizing errors, drawing statistical conclusions...), therefore, I advise the Authors to take it into consideration for the future work designs.

Author Response

The use of at least 3 technical and/or 3 biological replicates is highly recommended to improve the quality of the results (minimizing errors, drawing statistical conclusions...), therefore, I advise the Authors to take it into consideration for the future work designs.

We appreciate the reviewer's comment and will take these concerns into consideration when completing future studies.